

# Scales of collective entrainment and intermittent transport in collision-driven bed load

Dylan B. Lee[1] and Douglas Jerolmack[1]

[1]Earth and Environmental Science, University of Pennsylvania, 251 Hayden Hall, 240 S 33rd St, Philadelphia, PA 19104

**Correspondence:** Douglas Jerolmack (sediment@sas.upenn.edu)

**Abstract.** Fluvial bed-load transport is notoriously unpredictable, especially near the threshold of motion where stochastic fluctuations in sediment flux are large. A general statistical mechanics framework has been developed to formally average these fluctuations, and its application requires an intimate understanding of the probabilistic motion of individual particles. Laboratory and field observations suggest that particles are entrained collectively, but this behavior is not well resolved. Collective
entrainment introduces new length and time scales of correlation into probabilistic formulations of bed-load flux. We perform a series of experiments to directly quantify spatially-clustered movement of particles (i.e., collective motion), using a steep-slope 2D flume in which centimeter-scale marbles are fed at varying rates into a shallow and turbulent water flow. We observe that entrainment results exclusively from particle collisions and is generally collective, while particles deposit independently of each other. The size distribution of collective motion events is roughly exponential and constant across sediment feed rates.
The primary effect of changing feed rate is simply to change the entrainment frequency, although the relation between these two diverges from the expected linear form in the slowly-driven limit. The total displacement of all particles entrained in a collision event is proportional to the kinetic energy deposited into the bed by the impactor. The first-order picture that emerges is similar to generic avalanching dynamics in sandpiles: "avalanches" (collective entrainment events) of a characteristic size relax with a characteristic timescale regardless of feed rate, but the frequency of avalanches increases in proportion to the feed
rate. The transition from intermittent to continuous bed-load transport then results from the progressive merger of entrainment avalanches with increasing transport rate. As most bed-load transport occurs in the intermittent regime, the length scale of collective entrainment should be considered a fundamental addition to any probabilistic bed-load framework.

## Introduction

Bed load, the motion of particles along a stream bed by rolling, hopping and sliding, is the dominant mode of transport in rivers
for particles larger than 10mm (Parker et al. (2007); Dade and Friend (1998); Jerolmack and Brzinski (2010)). Bed-load flux equations lose their predictive power as fluid stress decreases toward the threshold of motion (Recking (2010)), where sediment transport becomes increasingly intermittent and exhibits fluctuations across a wide range of length and time scales (Ancey et al. (2008a); Singh et al. (2009); Ancey and Heyman (2014); Heyman et al. (2013)). Unfortunately, gravel-bed rivers organize their bankfull geometry such that they are always near threshold (Parker et al. (2007); Phillips and Jerolmack (2016)). There are two
potential causes of intermittency in near-threshold bed load: (i) variability in the driving stress due to turbulent eddies near the



bed (Nelson et al. (1995); Papanicolaou et al. (2001); Sumer et al. (2003); Diplas et al. (2008); Schmeeckle and Nelson (2003)); and (ii) variability in the resistive force of the bed due to structural arrangements of the grains (Charru et al. (2004); Martin et al. (2014); Prancevic and Lamb (2015); Yager et al. (2007)). While the role of turbulence has received the most attention, granular contributions to bed-load dynamics are increasingly being recognized (Frey and Church (2011a); Houssais et al.

(2015); Maurin et al. (2016)). One of the defining features of granular systems is a continuous transition from flowing to static regimes known as the jamming transition. On approach to jamming, particle motion becomes progressively slower and more heterogeneous; the variance in fluctuations of particle displacements grows rapidly (Keys et al. (2007); Liu and Nagel (2010)). Experiments show that the onset of bed-load transport has the hallmarks of a jamming transition (Houssais et al. (2015); Maurin et al. (2016); Houssais et al. (2016)). Near-threshold transport rates exhibit strong correlations and intermittency, while fluxes

at rates far above threshold are uncorrelated and smooth (Singh et al. (2009)). Since most gravel-bed rivers exhibit bank-full fluid stresses only marginally above threshold (Parker et al. (2007); Phillips and Jerolmack (2016)), this implies that these channels exist near the jammed state (Frey and Church (2011b); Houssais et al. (2016)). The stress distribution along a river bed is expected to exhibit a complicated structure, making the granular response to an applied fluid stress highly unpredictable (Albert et al. (2000)). Moreover, particle motion is expected to be highly localized and to exhibit nontrivial fluctuations and

hysteresis.

Given these challenges, and the many-body nature of the problem, one sensible approach is to examine bed-load transport in a probabilistic framework after Einstein (1950). Einstein defined a bed-load flux function of the form $q_x = E\bar{L_x}$. In this formulation of bed-load flux $q_x$, the entrainment rate function $E$ assumes a fixed timescale for the exchange of an inactive particle with an active one. More importantly, it assumes that both entrainment and deposition of particles are a time indepen-

dent, Poisson process and that particles do not interact. With these assumptions, the probability of entrainment is dependent only on flow conditions and the intensity of bed-load transport in an area of the bed. The entrainment probability can also be used to compute an average hop length $L_x$. The discussion above, however, indicates that bed-load transport has characteristics that deviate from the time-independent, non-correlated process assumed by Einstein. Indeed, experimental and field observations have revealed extreme fluctuations in particle activity/flux above the mean (i.e. extreme variance) (Gomez and Phillips

(1999); Nikora et al. (2002); Singh et al. (2009); Ancey et al. (2008a)), collective grain motion (Drake et al. (1988); Dinehart (1999); Ancey et al. (2008b)), and anomalous diffusion of particles (Ganti et al. (2010); Tucker and Bradley (2010); Phillips et al. (2013)). Starting with Ancey et al. (2008b), a series of models for bed-load transport have been proposed that posit that particles are often entrained collectively rather than individually at low mean transport rates. These models propose modifications of Einstein's entrainment function that take this correlated behavior into account through the introduction of a collective

entrainment rate, $\mu$, that leads to a characteristic correlation length scale, $l_c$ (Ancey et al. (2008b); Heyman et al. (2013); Ma et al. (2014); Heyman et al. (2014)). As the mean transport rate is lowered, modeled values for $\mu$ and $l_c$ must increase in order to reproduce the observed growth in variance of bed-load activity (i.e., the number density of moving grains). Collective entrainment is thus hypothesized to be the primary driver of observed intermittent and correlated bursts in bed-load transport near threshold; however, it has not been directly observed and quantified. Furbish et al. (2017) has taken a more generalized

approach to modeling stochastic bed-load transport, by viewing all probabilistic formulations of bed-load flux that incorporate



diffusivity as an approximation of a Master equation that exactly conserves both probability and mass. One key to making this approximation effective is a deep understanding of the underlying assumptions used to construct the effective diffusivity of the particles. While Einstein (1950) and others have assumed that bed load transport could be modeled as a Brownian process, there are important differences between bed load and Brownian motion. These can lead to major departures in how the diffusion

approximation is to be applied and interpreted in the context of bed-load transport. For example, recently Fathel et al. (2016) showed that the apparent anomalous behavior in the diffusivity of bed load particles at short times is actually a byproduct of the exponential growth in the variance of particle hop lengths as particle travel times are shortened. Furbish and colleagues' statistical mechanics framework is the most general model for bed-load transport; given knowledge of the microscopic and probabilistic motions of particles, one may derive continuum expressions for the macroscopic behavior. Collective particle

motion could be incorporated into this framework, but this requires an intimate understanding of the associated scales and correlations.

While the probabilistic approach has proven valuable for describing the nature of transport near threshold, it is vital to link this description to the physical origins of the stochastic behavior. If collective entrainment is the primary cause of bed load flux intermittency then what leads to it? One possible mechanism for collective motion is collisional impulses. Collisions are widely

recognized as drivers of bed load transport in aeolian systems where separate thresholds for entrainment without collision, the fluid threshold, and with collisions, the impact threshold, have been defined (Bagnold (1941); Martin and Kok (2016)). In aeolian systems these collisions are accompanied by dramatic 'splash' events where numerous particles are ejected at once (i.e. collectively). Recently, it has been proposed that entrainment in sub-aqueous systems has a significant collisional component as well especially in the case of large Stokes numbers (Pähtz and Durán (2017)). The Stokes number is the ratio of a particle's

inertial forces to the viscous forces of the fluid and, for binary collision between same-sized spheres, is given by (Schmeeckle et al. (2001)): $St = (\frac{1}{9})\frac{RDu_s}{\nu}$. Here, $R$ is the submerged specific density, $u_s$ is particle velocity, $\nu$ is fluid viscosity, and $D$ is particle diameter. For $St > 10^2$, viscous damping of collisions is negligible (Schmeeckle et al. (2001)) and thus collisions from saltation are expected to impart significant momentum to both the bed and neighboring particles for $D \geq 10^{-2}m$. Thus, it is likely that in coarse gravel streams, colliding particles cause a subdued 'splash' similar to aeolian systems. If the analogy with

aeolian systems holds then this splash entrainment will involve many particles becoming entrained at once. This hypothetical, collision-induced collective entrainment could be strong enough to be a primary driver of burstiness in bed-load flux near threshold.

There are other physical systems examined previously that organize themselves near a threshold, and display intermittent mass flux; the behavior of avalanching sand and rice piles comes to mind (Rajchenbach (1990); Lemieux and Durian (2000)).

These systems have been extensively studied and display intermittent transport in the limit where they are slowly driven past a threshold (in this case a critical angle). In the intermittent regime, the size and duration of avalanches is indeterminate (Frette et al. (1996)). As the sand pile is driven harder this intermittent regime gives way to continuous flow down the heap with an approximately constant flux. Hwa and Kardar (1992) showed how this transition into continuous flow can be viewed as a merger of the intermittent, avalanching events. Might bed load fit into a class of more generic avalanching systems that

transition from intermittent to continuous transport as they are taken from slowly driven to continuously driven?



In this paper we use the slowly to continuously driven limits as end members to explore how the nature of particle activity in an idealized bed-load experiment changes as the frequency of mean transport is varied. This is achieved by using a system that allows for precise control of the sediment feed rate while all other parameters (slope, fluid discharge, etc) are held constant, while particle motion is tracked using sequential images. The imposed feed rate is analogous to a driving frequency. We

replicate the previously observed growth in the intermittency of transport as the imposed sediment feed rate/driving frequency is slowed. Our major contribution is the direct observation of collective entrainment, which reveals that collisions release spatially-grouped clusters from the bed that are analogous to avalanches. We relate the scales of collective entrainment to the kinetic energy deposited into the bed by colliding saltators. This lends credence to the hypothesis that saltator-bed collisions play a large role in entrainment (both collective or otherwise). In our experiments, the growth in intermittency in bed-load

transport appears to arise primarily from the non-linear growth in the waiting times between transport events as the driving rate is slowed.

## Methods

### Experimental setup

The experiments are conducted using a narrow, quasi 2-dimensional (2D) flume in which all the grains in the subsurface and

surface can be monitored. The flume channel is 2.3 meters long and 20 mm wide. For all experiments, two different sized spherical glass beads, 12mm and 16mm in diameter, are fed into the channel in an even mixture. The two different sizes are chosen to ensure a randomly packed bed. The "quasi 2D" nature of the experiment arises from the fact that the small glass beads have significant overlap with one another along the axis orthogonal to the viewing window. All experiments are conducted in a flume slope of 6%, and a fixed discharge of 37.9 liters per minute, while the feed rate at which the particle mixture is introduced

to the channel is varied. The feed rate is the control parameter used in the experiments, and throughout the rest of the paper will be referred to as the driving frequency. The driving frequencies used for the experiment were: 40, 60, 80, 160, and 200 marbles per minute. Throughout the paper the abbreviation MPM will be used for marbles per minute.

At the flow rate used, all flows in the channel are turbulent with Reynolds numbers greater than $10^4$. Flow depths were found to be uniform with the exception of 10-15 cm near the inlet and outlet of the flume. The flow is supercritical with Froude

numbers greater than one, though any bedforms that would be present at these flow conditions are suppressed due to the narrowness of the channel. Experiments are in the high Stokes number regime where collisions are expected to be important, in order to mimic the conditions of gravel-bed rivers. Although collision velocities vary (they are quantified below), they scale roughly with settling velocity; using terminal fall velocity as a scale parameter, $St > 10^2$ for all experiments. [] Details about the flow parameters observed during the experiments are given in table 1. Only mean flow parameters are listed as the flow

parameters are kept approximately constant across experiments. This was verified during the experimental runs where the range of flow conditions that occurred during a single experiment was similar to the variability in conditions seen across experiments. This flume is thought to represent the simplest possible physical model of bed-load transport. A diagram of the experimental



Earth **Surface**
**Dynamics**
Discussions



**Figure 1.** Schematic of the experimental setup. The system is 2.3 meters long and 20 mm wide. This quasi-two dimensional channel is fed at a constant water discharge for all experiments. The slope is kept constant at 6 %. The sediment feed is uncoupled from the fluid discharge, and is introduced from above using a custom designed feeder built at the PennSed laboratory. A viewing window on the order of 35 cm is selected two thirds of the way down the flume. The window is back-lit and the resulting images can be seen in the figure inset.





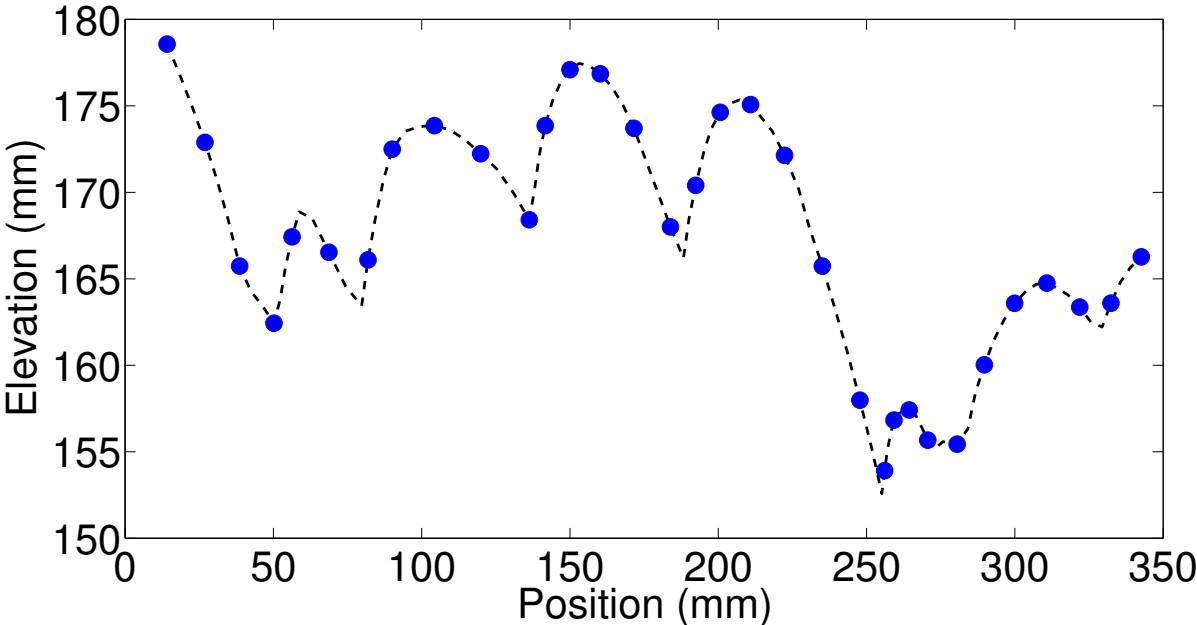

**Figure 2.** An example saltator trajectory obtained during one of the experimental runs. Trajectories are created for all particles present in the sampling area of the experiment.

setup can be seen in figure 1. The experiment is very similar to that used by Ancey (Ancey et al. (2008b)). This is intentional so that their results can guide our study and our findings can be compared to their data.

A camera is situated approximately 100 cm downstream of the flume inlet. The viewing window of the camera is 35 cm for all experiments. This section of the flume is back-lit using a white LED panel array that outputs at 300 lumen. This produces a

5 sharp silhouette of all the grains in the viewing window that can then be used to acquire approximate particle centers. Images are acquired at a rate of 120 fps and streamed continuously to a computer. This acquisition rate is necessary to adequately capture the trajectories of individual particles as they move through the viewing window.

**Data acquisition and analysis**

Once images are acquired, approximate particle centers are located using a hybrid form of the algorithms outlined in Khan and

10 Maruf (2013) and Parthasarathy (2012). Using this method it is possible to obtain particle centers that are accurate to better than 1 mm. However, the method is highly sensitive to the degree of occlusion that the particles in the bed experience, and thus centers can sometimes be less accurate. Once particle centers are obtained, particles are linked together from image to image



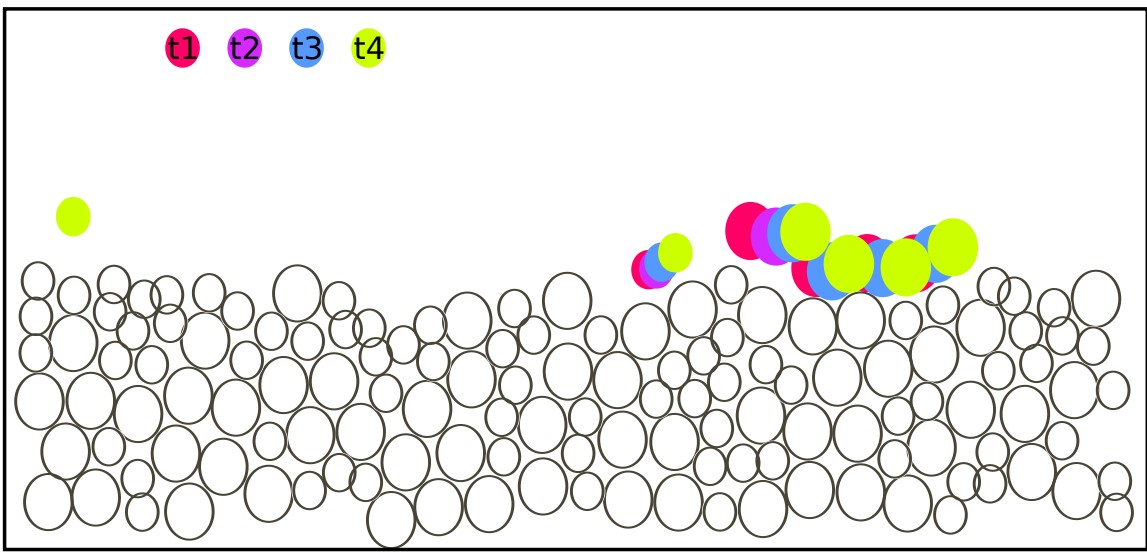

**Figure 3.** An example of a collective motion event sampled at 4 different times during the event. The particles that are displacing actively during the event are color coded according to their position at the time step (t1, t2, etc) associated with a given color. At t1 only two particles are moving. The large particle collides with three particles on the bed at t2 and these three particles displace at t3 and t4. The four large particles would be classified as moving together collectively.

| Experimental flow parameters | | | | | | |
|---|---|---|---|---|---|---|
| h (mm) | slope(%) | $\bar{u}_f$(m/s) | *Sh* | *Fr* | *Re* | *St* |
| 33.6 | 8.5 | .73 | .14 | 1.3 | 10.7E3 | >1000 |

**Table 1.** Mean flow conditions observed during the experiments. All values are means taken around the range of flow conditions observed over all experiments. h is the mean flow depth, $\bar{u}_f$ is the mean flow velocity, *Sh* is the Shields number, *Fr* is the Froude number, *Re* is the Reynolds number, and *St* is the Stokes number of the large diameter grains. *St* for the small diameter grains is also much larger than the viscously damped limit.



to obtain particle trajectories using the method outlined by Crocker and Grier (Crocker et al. (1996)). An example trajectory that is the final output of this process can be seen in figure 2.

With a particle trajectory affixed to each particle that enters and leaves the viewing window, it becomes possible to analyze the dynamics of mobile particles over a wide variety of timescales. Emigration events sampled in the viewing window are also
easy to obtain from these trajectories. Emigration series are obtained by choosing a fixed along-stream distance, $x$, to sample along the viewing window for all experiments in question. If a particle center crosses this position in the downstream direction it is counted as a positive emigration event. If it crosses this position in the upstream direction it is counted as a negative emigration event. This definition is identical to that used in Ma et al. (2014). To study the active particles within the viewing window it was necessary to set a threshold for particle mobility. To do this particle trajectories were analyzed over 1/10 of a
second. If the particle displaced 2.4 mm within this time window then the particle was deemed mobile.

One approach to estimating the intermittency of a time series is to look at how long one needs to sample to arrive at a threshold standard deviation. In the case of a uniform, low intermittency time series this sampling timescale will be very short, whereas in the case of a highly intermittent series a long sampling time will be needed. This timescale is referred to throughout the rest of the paper as $t_{conv}$. It is computed directly from the obtained emigration series for all of the driving frequencies
studied, by incrementally increasing the time, $\tau$, used to sample from the emigration series. For a given $\tau$, 500 samples are randomly selected from the emigration series in question using a Monte Carlo technique. The standard deviation of these samples is then computed and normalized by the mean emigration rate for the samples taken from the series. As $\tau$ grows, the standard deviation approaches the value chosen as the threshold standard deviation, $t_{conv}$. When the threshold standard deviation is reached this value is interpreted as $t_{conv}$. This approach is identical to that used in Houssais et al. (2015).

Waiting times are sampled from the emigration series as well. They are interpreted to be the time periods in between active emigration events over position $x$. A waiting time period is started after an emigration event over position $x$ occurs, and it ends when the next emigration event happens.

Activity within the whole viewing window sampled in the experiments is characterized in the paper through two different event-based metrics. One type of event is referred to specifically as a "collective entrainment event". This is defined as a group
of one or more mobile particles (mobility was determined using the criteria above) moving within one large-grain diameter of each other. This analysis is a simplified version of that used to identify mobile clusters in Keys et al. (2007). An example of a collective entrainment event is given in figure 3. In this example, the 4 large grains that are in color would be considered a collective entrainment event. Collective entrainment events were identified directly from analysis of the mobile particle trajectories sampled in the viewing window for a given experiment. For a given time step all $N$ mobile grain trajectories were
identified. For $i = 1$ to $i = N$, the distance of the $i$th mobile grain to all the rest of the mobile grains was computed. A clustering algorithm was then employed to identify clusters of grains that were within a threshold distance of one another. This algorithm is capable of identifying an arbitrary number of mobile clusters occuring at the same time within the viewing window. A single mobile cluster of grains is defined as a collective entrainment event. This cluster analysis was performed for the entirety of the time steps available for a given experiment. This allowed us to gather statistics of all of the collective entrainment events that
occurred in the viewing window for a given experiment.





The other type of event is a "transport event"; it is more general, and contains collective entrainment events within it as a subset. It is defined as a time period where there is at least one mobile grain within the viewing window. As long as this is the case, an event is said to be taking place. Once there are no mobile grains within the viewing window then the transport event has stopped. A portion of a transport event is pictured in figure 3. Here all the grains that are colored are considered to be

part of the current transport event that is taking place. It is possible to see there are time instances in figure 3 where collective motion is not occurring but particle activity is still ongoing. These time instances with no collectively moving particles would be counted as part of a transport event, but not as part of a collective motion event.

To analyze the effects of impacts during events, saltating grains were separated from the rest of the mobile population for a given event. The trajectories of the saltators were then numerically differentiated twice to obtain acceleration series

of the trajectories. A change-point detection algorithm was then employed to identify the spikes in the acceleration series representative of impact events.

## Results

All of our experiments exhibited intermittent particle activity (figure 4). This intermittency confounds efforts to determine the time needed to arrive at an average activity or flux for a given rate of transport, $\Delta t_{conv}$. We computed $\Delta t_{conv}$ for all

driving rates used in the present study, and found that it declines monotonically with increasing feed rate (figure 5A). A naive expectation for the decrease in the averaging time is that it should be proportional to the inverse of the driving frequency; that is, $\Delta t_{conv} = a/Q_i$, where $a$ is a scaling parameter that depends on the percent standard deviation threshold chosen to determine $t_{conv}$. This relation can be thought of as marking the growth in time required to count a fixed number of particles emigrating past a line if driving frequency were the only thing that mattered. This should be the case at high transport rates

where we expect smoother transport; accordingly, we choose $a = 110$ such that the relation matches the observed data for the highest feed-rate experiment. We see that the naive relation describes the convergence time reasonably well for the three highest feed-rate experiments (figure 5). For the two slowest feed rates in the study, however, the actual increase in averaging time with a reduction in $Q_i$ is more rapid than this expectation.

The waiting times between all observed emigration events for a given experiment were sampled, and used to compute

empirical complementary cumulative distributions (figure 6A). We compare these distributions to a Poisson distribution with a value $\lambda = 1$, which is on the order of the mean waiting times seen in the experiments. The Poisson distribution was chosen for comparison because of the extensive body of literature showing its fitness for modeling uncorrelated random processes (Lawler and Limic (2010)). If the waiting times are uncorrelated and truly random they should follow a Poisson distribution; however, the measured waiting times decay much more slowly (figure 6A). As expected, the waiting times between emigration events

seem to be a function of the driving frequency. When the former are non-dimensionalized by the latter, the variance among the experiments is significantly reduced (figure 6B). Moderate dispersion remains among the different experiments, however, indicating that driving rate is not the only factor controlling the waiting times. We compute the average waiting time for each experiment; the naive expectation is that this waiting time is precisely the inverse of the driving frequency. The data follow this





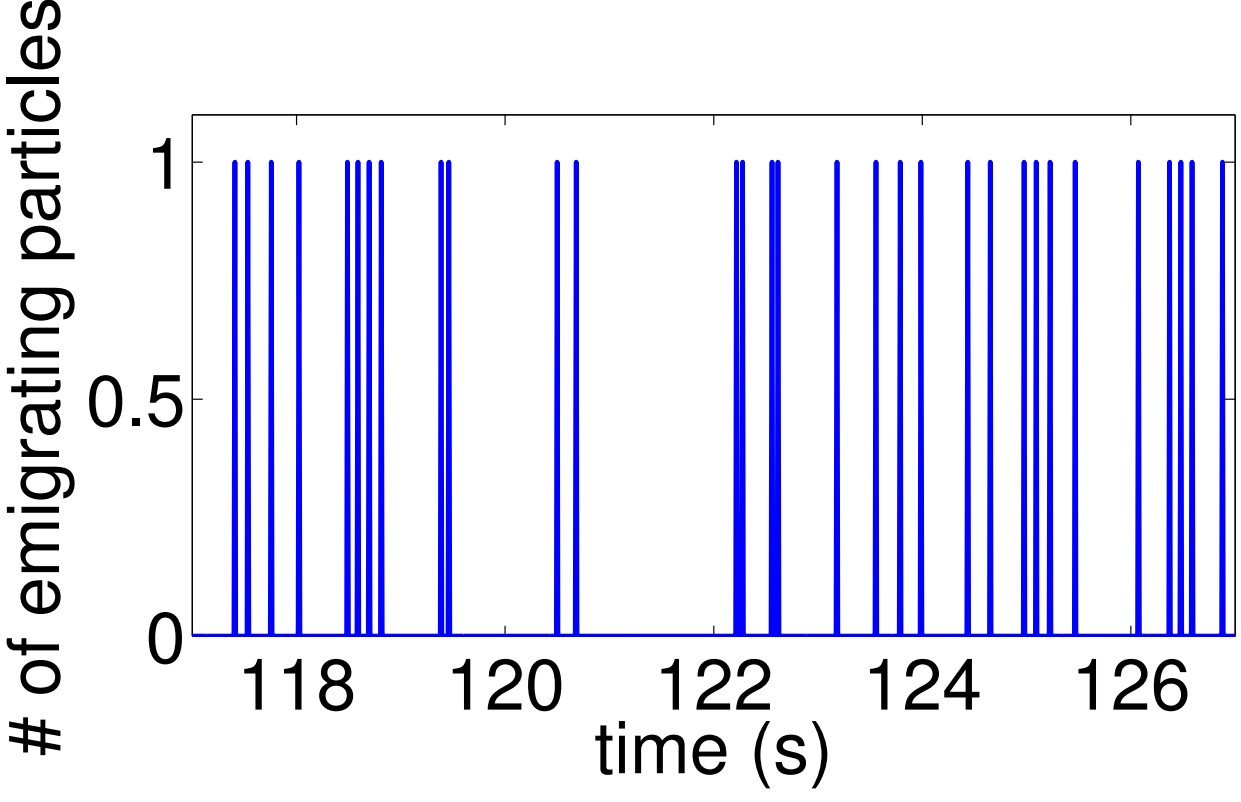

**Figure 4.** Example time series of emigration events for $Q_i = 40$ MPM. A position $x$ along the bed viewing window (as seen in Figure 1) is monitored during the experimental run. When a particle passes position $x$ it is considered an emigration event. This is a simple measure of particle activity that can be converted to a time-averaged flux. Time series of emigration sampled at a fixed position $x$ along the bed were determined for all experimental runs.

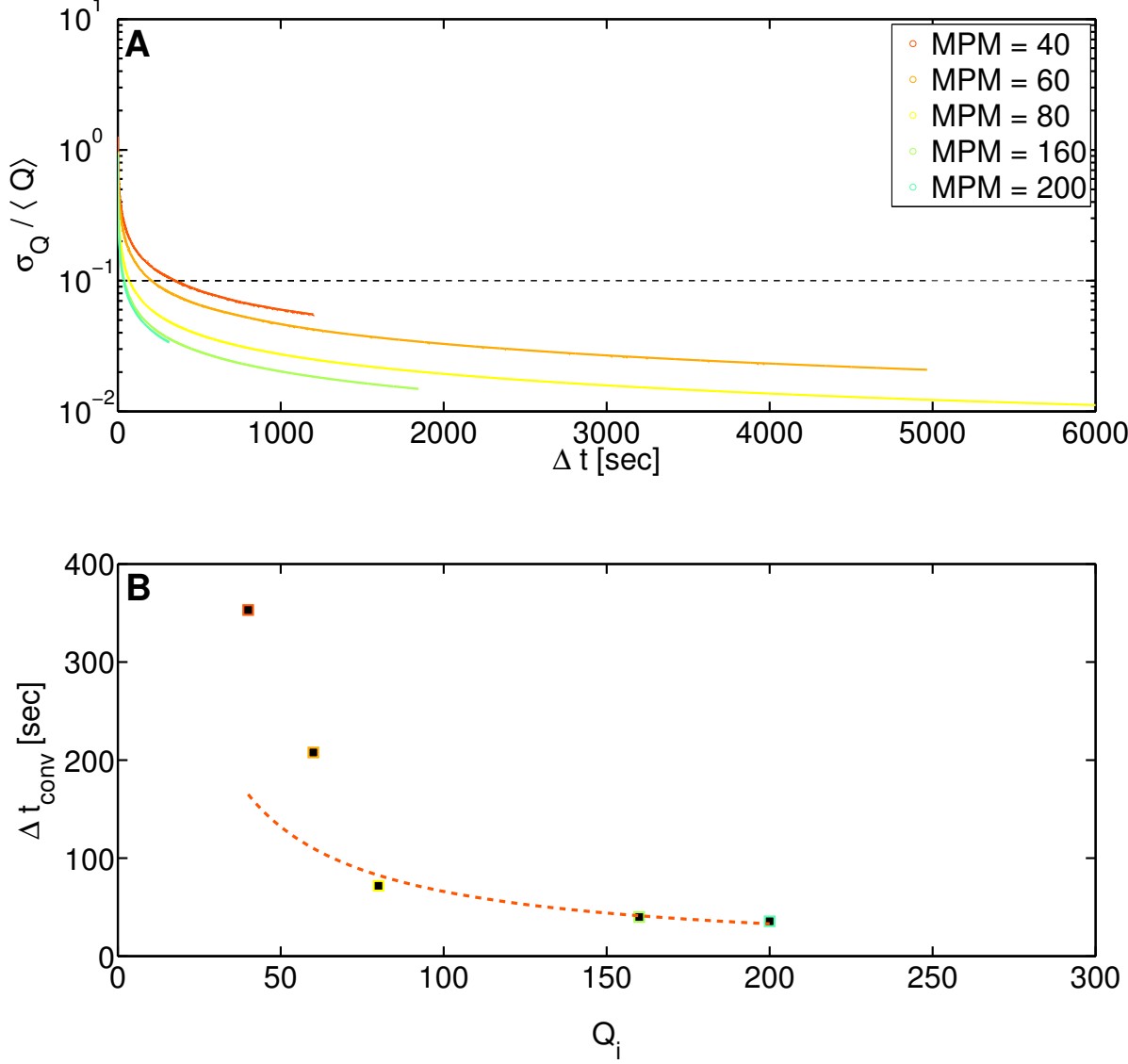

**Figure 5.** Determination of convergence time for experiments at all driving rates. (A) Standard deviation of an ensemble of samples over a given $\Delta t$. As $\Delta t$ grows the the standard deviation decreases and approaches the threshold standard deviation. This value of $\Delta t$ is interpreted to be the convergence time $t_{conv}$. The standard deviation is normalized by the mean emigration rate Q. Legend indicates feed rate in marbles per minute (MPM). (B) The time $\Delta t_{conv}$ necessary for flux measurements to converge to a threshold standard deviation of 10 percent, as a function of the driving frequency in number of marbles per minute. The dotted markers are the actual observed convergence times, while the dashed red line displays the trend that one would expect the convergence time to take if it were simply a function of the feed frequency ($\Delta t_{conv} = 110/f_{input}$; see text for details).





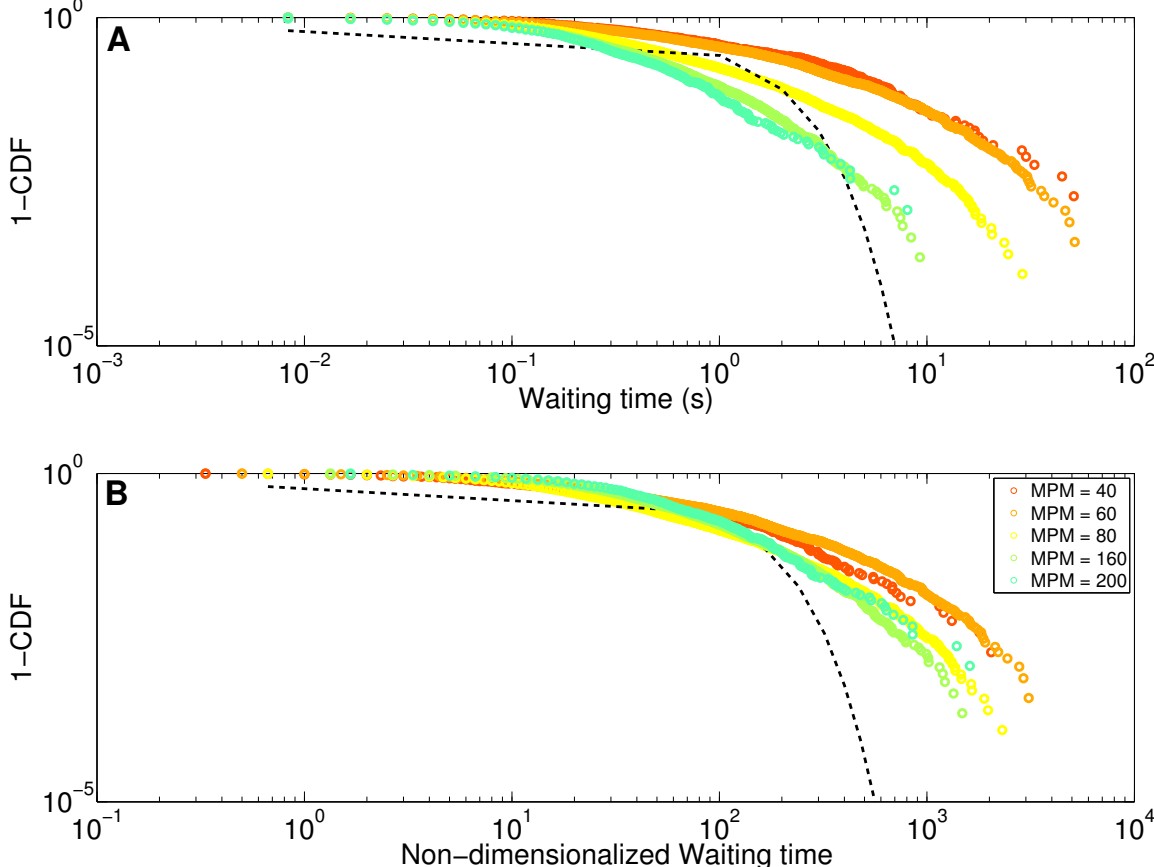

**Figure 6.** Complementary cumulative probability plots of waiting times between emigration events; (A) data for all experiments, and (B) the same data normalized by the driving frequency of each respective experiment. Expectation from a Poisson distribution is shown for comparison with dashed line. Legend as in figure 5.

expectation for the three highest driving frequencies; however, the mean waiting times for the two slowest-driven experiments are significantly larger than expected (figure 7).

The above results demonstrate that driving frequency has a strong effect on the timing of emigration events, and the timescale required for averaging. To determine if this frequency also effects how particles are transported, it is necessary to look at the particle kinematics during times when particle activity is present. We examine here the complementary cumulative distributions of particle speed for all experiments (figure 8). The tails of the speed distributions do not vary strongly as a function of driving frequency. This is sensible as the slow speeds ($< 0.1$ mm/s) are associated with essentially immobile particles, while the fast speeds ($> 100$ mm/s) are almost exclusively associated with saltators. As the fluid discharge is kept constant across



**Figure 7.** Relationship between the driving frequency $f_{input}$, and mean waiting time between observed emigration events $W$, for each experimental run. Dashed red line shows the expected relation that the mean waiting time is the inverse of the driving frequency, $W = 1/f_{input}$.



**Figure 8.** Complementary cumulative distributions of active particle speeds for all experiments. Legend as in figure 5.

experiments, we do not expect to see large differences in the speed of saltating grains. We do see an effect of driving frequency, however, for the intermediate speeds (figure 8). As the driving frequency declines, the transition between mobile (fast) and immobile (slow) particles appears to grow more abrupt; this is manifest as a growing kink in curves. In other words, the distribution of particle speeds is more continuous at high driving frequencies, and becomes more bimodal at low driving

5  frequencies as motion separates more distinctly into slow and fast particles.

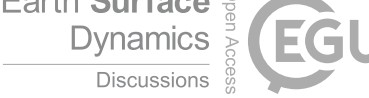



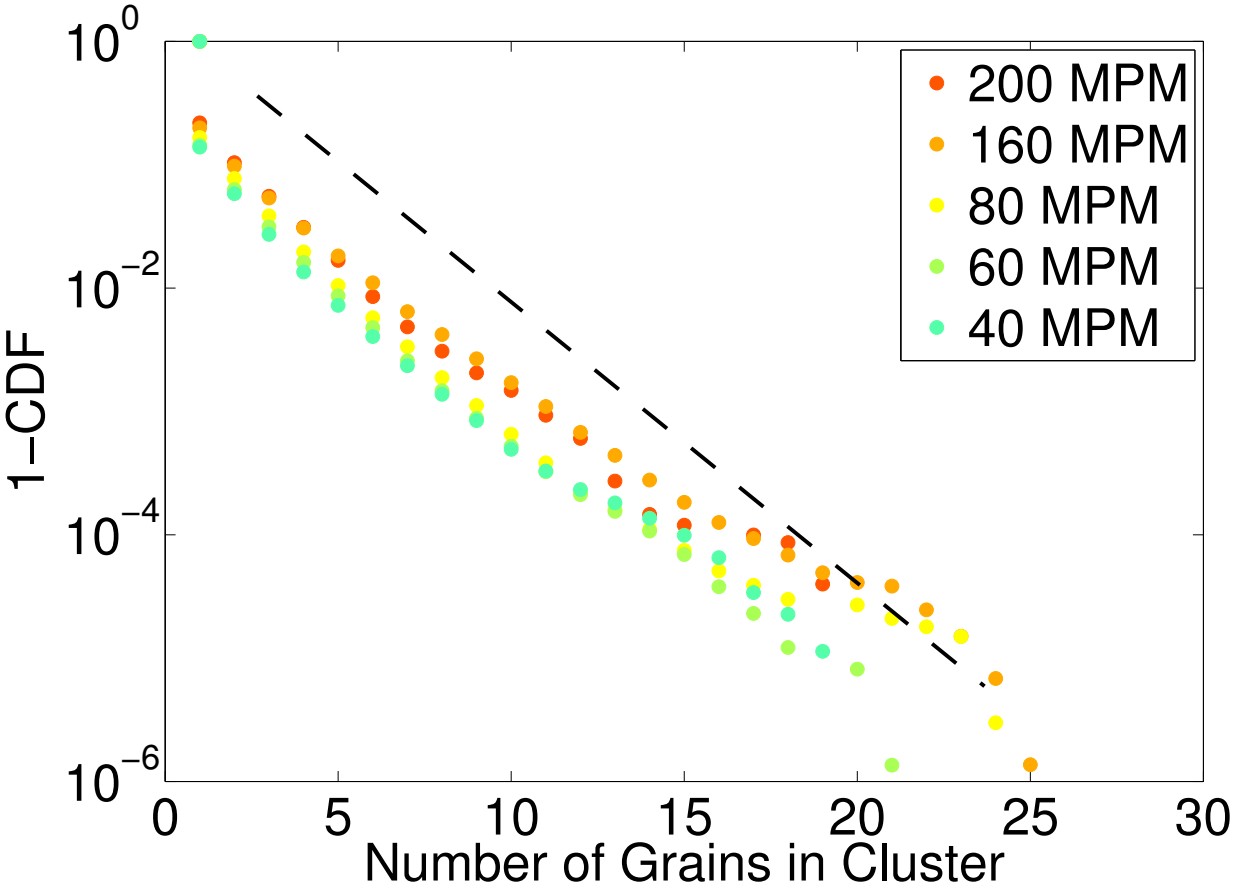

**Figure 9.** Cumulative complementary distributions of mobile cluster sizes, for each experiment at a different driving frequency. A cluster is defined as a group of mobile particles moving together in space. The dashed black line is an exponential trend, plotted for the sake of comparison. Note logarithmic y axis.

Thus far we have examined the motion of individual particles. Here we consider collective entrainment events — in particular, the size distribution of particles that have been determined to be moving together. These mobile clusters are analogous to avalanches in other granular systems. Interestingly, the distribution of mobile clusters does not vary significantly with driving frequency (figure 9). All experiments show a roughly exponential distribution of cluster sizes, with a mean size that varies only slightly with driving frequency.

We observe qualitatively that almost all entrainment is associated with impacts. However, the exact nature of this relationship is extremely difficult to untangle for individual entrainment events. Entrainment can happen immediately after an event, or after an unpredictable time delay. In addition, because the disordered bed absorbs and transmits momentum in a complex way, a particle can be entrained as a result of an impact that happened a significant distance ($>> D$) upstream. To avoid these issues,





while still gaining insight on the effects of impacts on entrainment, an attempt to look at all impacts for a given period of particle activity in the observational window of an experiment was performed. An event was defined as a period where at least one particle was always mobile. Once all particles in the observation window become immobilized, the event is deemed over (see above). For a given event we computed: (i) the amount of kinetic energy (KE) deposited into the observed section of the bed; (ii) the cumulative displacement of all mobilized particles; and (iii) the number of particles mobilized. Deposited KE was determined by identifying the points in time when an entrained saltator collided with the bed. The saltator velocity immediately before the collision and the velocity immediately after the collision was used to obtain the difference in KE of the saltating particle that occurs as a result of the collision. This difference in KE was interpreted as occurring because of the inelastic collision of the saltator with the bed, and can be interpreted as being the kinetic energy transferred (or deposited) by the saltator into the bed. The KE deposited, cumulative displacement, and the number of particles mobilized was compiled for all events and for all driving rates, in order to determine the extent to which particle mobility may be understood from collision energetics. The data reveal a remarkably clear, linear relation between the total KE deposited and the cumulative displacement of mobile particles (figure 10A). We also see that the number of mobilized particles systematically increases with KE deposited, though there is significant variability (figure 10B).

**Discussion and conclusion**

For all driving frequencies, both the magnitude of collective entrainment events (figure 9) and the speed of saltating (fast) particles (figure 8) are similar. This indicates that collision dynamics do not vary significantly across the range of sediment feed rates probed here. Roughly, the intermittency of transport is controlled by the growth in the mean waiting time as the driving frequency is slowed (see figure 7). Changing the driving rate appears to primarily affect how quickly events happen, and not the fundamental nature of entrainment. In the slowly driven limit, (collective) entrainment events are infrequent and may be considered as discrete bursts in transport. As the system is driven harder, events occur more frequently and begin to overlap with one another. At the fastest driving rates, events become indistinguishable from one another and continuous transport emerges. This picture aligns with behavior seen in avalanching systems that display an intermittent to continuous transition (Hwa and Kardar (1992); Rajchenbach (1990)). A sand pile model by Hwa and Kardar (1992) showed that overlapping avalanches may interact, introducing correlations in the flux output of the system. The observed changing distribution of particle speeds with driving rate in our experiments may be an indication of this kind of complex behavior.

While much of the difference in entrainment rate and intermittency can be related simply to the driving rate, some of it cannot. In particular, at low driving rates we see waiting and averaging times that are significantly longer than expected, suggesting that the first-order kinematic avalanching model described above is incomplete. One timescale that has not been considered is the relaxation time of avalanches, which for our experiments would be the deposition timescale of mobile clusters following an entrainment event. This timescale may not be independent of driving rate, and it becomes impossible to measure when avalanches overlap in time. Another complicating factor at low driving rates is the influence of creep, which has been

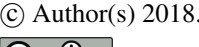



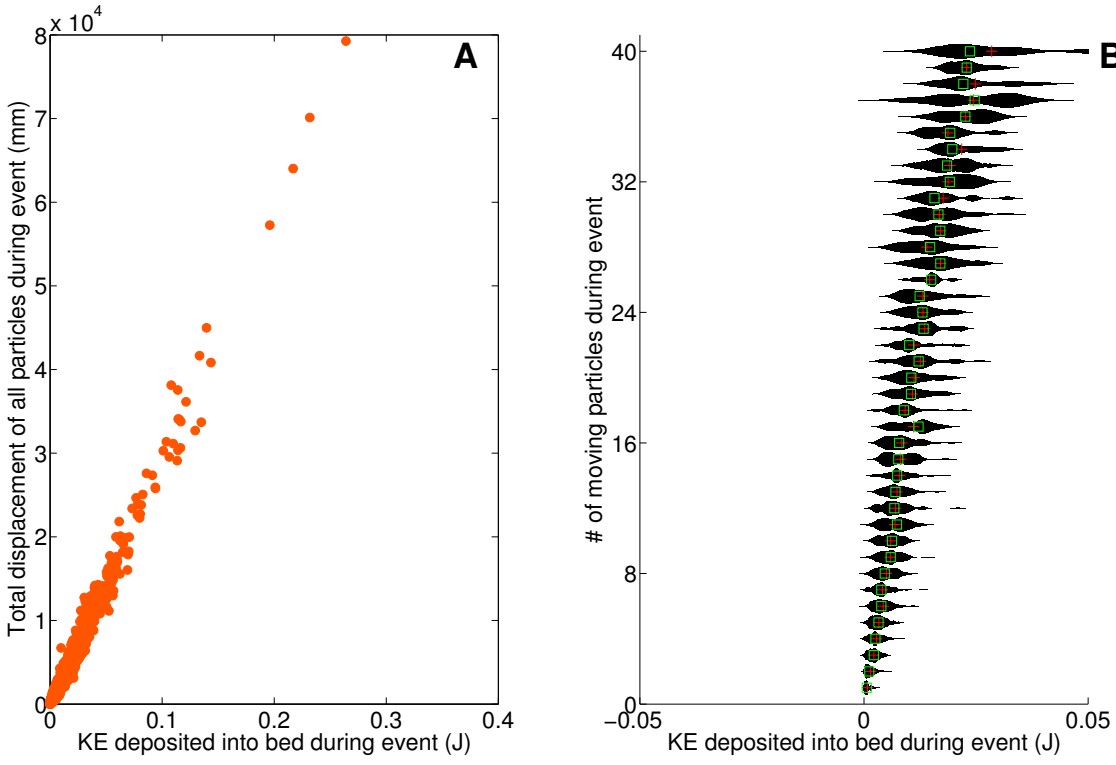

**Figure 10.** Particle mobility as a function of kinetic energy (KE) deposited into the bed for an event. (A) Cumulative displacement for all mobile particles during an event increases linearly with KE deposited. (B) Probability distribution of the number of mobile particles, as a function of the KE deposited. The mean of the distribution is displayed as a red cross, and the medians are shown as green squares.

demonstrated to drive bursty bed-load transport in the viscous flow regime (Houssais et al. (2015)). Movies of our experiments reveal the presence of slow creep also, but quantifying its significance is beyond the scope of the present paper.

Indeterminate, complex behavior (such as the possible scenario outlined above) is often an inherent feature of many-bodied, driven and strongly dissipative systems (Regev et al. (2013)). For our system of a turbulent fluid driving marbles that collide

5   with a bed, it is not possible to predict the response of a collision. Some collisions result in a strong rebound of the saltator and no (observable) response of the bed; others drive an immediate splash as several grains are entrained; and yet others lead to a delayed response, in which a large number of grains become destabilized and slowly accelerate to become entrained. Knowledge of the kinetic energy of an impact is not sufficient to understand entrainment, due to the complicated nature of energy dissipation. Knowledge of energy dissipation, however, allows for significant predictive power. The strong relations

10  between energy deposited, and the size and cumulative displacement of entrained particles, provide some mechanistic basis for understanding collective entrainment and burstiness in collision-driven bed load.



The similarity of collective entrainment events between driving rates shows that, while collective entrainment is present, its associated length scale does not vary as a function of intermittency. While more analysis remains to be done, it is likely that collision-induced momentum transfer into the bed is what sets the scale of collective entrainment. Since fluid discharge did not vary in our experiments, the velocity of saltating grains (and hence impact energy) remained roughly constant for all

driving rates. The approximately constant exponential trend (figure 9) aligns with the expectation that momentum transfer due to saltator-bed impacts should be a primary driver of entrainment in this system. Ancey et al. (2008b) were correct in positing a length scale for collective entrainment; we see definite evidence for a length scale of particle motion that is larger than that of a single particle. While this length scale does not vary in these experiments, it is challenging to extrapolate to other systems. At smaller Stokes numbers, collisions are damped and turbulence becomes an important driver of collective entrainment (Nelson

et al. (1995); Papanicolaou et al. (2001); Sumer et al. (2003); Diplas et al. (2008); Schmeeckle and Nelson (2003)). The shapes and size distributions of natural particles, and roughness of the river bed (e.g., bed forms), will also likely influence collective entrainment in ways that are difficult to anticipate. Nevertheless, collision-driven entrainment should be the norm for gravel-bed rivers (Jerolmack and Brzinski (2010)), and collective entrainment has already been observed in the field (Drake et al. (1988)). Incorporating this length scale into the general probabilistic framework proposed by Furbish (e.g. Furbish et al. (2012, 2017))

will be important in the effort to build statistical-mechanical models of bed-load transport, that start with correct assumptions of the underlying dynamics that govern bed-load particle trajectories. Understanding the scales of bursty bed-load transport will also inform the requisite times for bed-load sampling in the field and laboratory (Singh et al. (2009)).

*Competing interests.*  The authors declare that there are no competing interests

*Acknowledgements.*  We thank Carlos Ortiz and Morgane Houssais for discussion and help with data analysis. This research was partially

supported by the US National Science Foundation (NSF) grant EAR-1224943 and the US Army Research Office, Division of Earth Materials and Processes grant 64455EV.




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
