# Peer review of "Determining the scales of collective entrainment in collision-driven bed load"

_Earth Surface Dynamics, 2018_

## Short Comment (SC1) · 23 Feb 2018

Dear authors,

I am not an official reviewer. However, because the findings in your paper seem to support major hypotheses of two of my recent papers,

[1] Pähtz & Durán (PR Fluids, 2017, doi: 10.1103/PhysRevFluids.2.074303)
[2] Pähtz & Durán (https://arxiv.org/abs/1602.07079) (this paper is still under review),

[Figure]

I decided to give you review-like comments, which I hope you may consider in revisions of your paper.

**Entrainment by particle-bed impacts (page 3, lines 15-25)**

1. Reference [1] is not the only study reporting a crucial role of impact entrainment. Vowinckel et al. (JHR, 2016, doi: 10.1080/00221686.2016.1140683) carried out DNS/DEM simulations and found that the probability of entrainment by turbulent sweep events is strongly increased when a particle-bed collision preceded the event.

2. In contrast to what your writing seems to suggest, the results we reported in Ref. [1] are independent of lubrication forces (and thus the Stokes number) in the case of bedload transport. In fact, we found nearly identical behavior for simulations with restitution coefficients $e = 0.9$ (no viscous damping) and $e = 0.01$ (nearly maximal viscous damping). We also found that impact entrainment dominates entrainment by the mean turbulent flow for sufficiently large 'impact number' $\mathrm{Im} = \sqrt{R + 0.5}\sqrt{(R-1)gD^3}/\nu$. I guess you may have been mislead by one of our statements in Ref. [1], where we mention that the impact number may be interpreted as a Stokes-like number. However, this statement does not mean that the impact number is the same as the Stokes number.

**Influence of feeding rate (Figs. 5B, 7, and 8; and page 16, line 27 and following)**

The results in Fig. 8, namely the bimodal distribution at low feeding rates and more continuous distribution at large feeding rates, are very similar to those we reported in Ref. [2] (e.g., see Fig. 9 in Ref. [2]) and seem to support the following hypothesis: We hypothesized that the probability that bed surface particles are entrained by an impactor and subsequently acquire an energy sufficiently high to

become a saltator depends on the impact frequency. To understand the background of this hypothesis, one can think of two impactors hitting a bed surface particle in short sequence. In this situation, the second impact obviously has a higher probability of entraining the bed surface particle and promoting it to a saltator provided the bed surface particle does not fully recover from the first impact. More generally, we argued that the larger the impact frequency the larger the creeping and fluctuation motion of the bed surface (as bed surface particles do not fully recover between particle-bed impacts), which weakens the links between neighboring bed surface particles and thus makes them more susceptible to receiving momentum from an impactor (associated with an increased probability of entrainment and promotion to saltators). At low feeding rates (i.e., low impact frequency), bed surface particles are strongly linked with each other and thus become very seldom saltators (hence, the bimodal distribution), whereas at large feeding rates (i.e., high impact frequency), bed surface particles are weakly linked and thus are readily promoted to saltators (hence, the more continuous distribution). We further argued that the characteristic creep and fluctuation velocity of bed surface particles reaches a critical value $\sim \sqrt{(R-1)gD/(R+0.5)}$, at which bed surface particles are at the brink of being mobilized (the weakest possible link to their neighbors), when a critical impact frequency is exceeded. A further increase of the impact frequency then does not anymore weaken the link between bed surface particles, and thus does not anymore increase the probability of impact entrainment and subsequent promotion to saltators by single impactors; only the trivial proportionality of overall impact entrainment to the number of transported particles remains. This hypothesis is consistent with Figs. 5B and 7: at large feeding rates the statistics of single particle-bed impacts are independent of the feeding rate and thus one expects the trivial proportionality to $1/f_{\text{input}}$; whereas at low feeding rates the probability of entrainment by single particle-bed impacts increases with the feeding rate and one thus expects a upward-deviation from the trivial proportionality to $1/f_{\text{input}}$. Hence, this hypothesis may complete what you call an incomplete picture in page 16, line 27 and following.
**ESurfD**
I believe your paper would strongly benefit from a discussion like the one I outlined above because the idea that particle-bed impacts are dominating entrainment in bedload transport is very new and often not taken seriously in the community (as I have experienced numerous times myself). Showing that experimental results support a hypothesized mechanism that had been previously suggested by numerical simulations from a different research group would add a very strong argument in favor of the impact entrainment idea.

Thomas Pähtz

---

## Referee Comment (RC1) · D. J. Furbish (Referee) · 1 Mar 2018

Review of:
Scales of collective entrainment and intermittent transport in collision-driven bed load
Dylan B. Lee and Douglas Jerolmack

**General Comments**

As Lee and Jerolmack point out, the recent work of Ancey, Heyman and others concerning collective entrainment represents an important step forward in focusing our attention on this problem as a significant part of the behavior of gravel-bed rivers, particularly near threshold. The innovative birth-death formulation of entrainment and deposition developed and elaborated by Ancey and others provides a natural probabilistic lens for highlighting the possible significance of collective entrainment. In turn, the work reported by Lee and Jerolmack in this paper represents a natural experimental extension of the idea of collective entrainment, with a focus on particle collisions, in terms of how this process contributes both to entrainment and to the intermittency of transport.

A straightforward, and clever, element of the experimental design is that transport begins only with the introduction of particles to the flow. This in principle isolates the effect of collective entrainment by collisions, rather than needing to parse these events from entrainment by fluid motions (a particularly difficult experimental problem) — although the flow certainly contributes to the continuing motions of introduced particles, and of particles following their entrainment by collisions; and it also likely contributes to entrainment of particles previously destabilized by collision events. The set up allows for direct observation of collective entrainment by collision. The ideas concerning the release of clusters similar to avalanching is particularly interesting, and the analysis concerning the relation between mobilized particles and the extraction of kinetic energy during particle collisions highlights the importance of examining this problem probabilistically.

The comments and questions below are aimed at seeking clarification of certain elements of the presentation, in part because the results likely will be used by others, and in part because I am curious about aspects of the interpretation.

**Specific Items**

**Page 1, Line 23:** Although I appreciate the intention of the phrasing, I suggest rewording, or just deleting the word "unfortunately."

**Page 2, Line 15:** The idea of "hysteresis" used here is unclear. Is this point needed?

**Page 2, Line 21:** Is it correct that the mean hop distance can be calculated from the entrainment probability?

**Page 2, Line 32:** Does the collective entrainment rate increase, or just its relative contribution, as the transport rate decreases?

**Page 3, Line 7:** I recommend replacing "exponential" with nonlinear.

**Page 3, Line 9:** I recommend replacing "continuum" with "continuum-like," as the rarefied conditions involved in this problem do not satisfy the continuum assumption. The referenced expressions are continuum-like only in that they have the continuous form of continuum equations, when in fact they pertain to ensemble expected conditions.

**Page 4, Line 28:** typo present []

**Page 8, Lines 11–19:** It is unclear what quantity is being calculated. (I gather later that this is either the activity or the flux, normalized by the average?) What is the total time series length from which the 500 samples are drawn (for specified $\tau$)? Do these samples overlap (representing the possibility of non-independence)? How might this algorithm differ from calculating the quantity of interest directly from the series with successively increasing duration $\tau$ (a standard approach for examining convergence)?

**Page 9, Line 9:** Were these identified acceleration spikes used in a specific manner later? For example, in the later calculations of kinetic energy changes? If so, are these component accelerations, or computed as changes in trajectory resolved speeds? At 120 fps, I can readily imagine that the acceleration time series are quite noisy, such that identifying impacts involves significant uncertainty?

**Page 9, Line 25:** The appropriate comparison is with an exponential distribution rather than a Poisson distribution. The latter is a discrete distribution of the (integer) number of events occurring within a specified interval, assuming a stationary Poisson process. The exponential distribution describes the wait times between successive events, assuming a stationary Poisson process. (Is this why the dashed line appears to segmented parts in Figure 6? Where the kinks in the line coincide with integer values $\geq 10^0$ and the left part of the line is extrapolate toward zero?)

Certainly the data (if expressed in terms of numbers of events) could be compared with a Poisson distribution using the average number of emigration events occurring during a fixed interval — if it is demonstrated that this number converges to a finite value, if the number of events is large, and if the probability of the occurrence of an event in the fixed interval is sufficiently small. If these conditions are not satisfied, then a comparison with the binomial distribution is preferable.

In any case, the key point that is being examined is whether the emigration events can be treated as being a Poisson process (whether viewed in terms of the waiting times, or in terms of numbers of events per specified interval), or whether the data in fact suggest absence of randomness in emigration events related to the collective entrainment process upstream. The exceedence probability of an exponential distribution plots as a line that curves downward in log-log space (Figure 1), as the data do in Figure 6. (Might I suggest also exploring a semi-log plot as is done late with respect to cluster sizes in Figure 9.)?

[Figure]

Figure 1: Plot of the exceedence probability function associated with an exponential distribution of waiting times for three values of the average waiting time.

When I use three of the average waiting times from Figure 7 (including the smallest and the largest) and compare this figure with Figure 6, my sense is that the author's conclusions stand, that an exponential exceedence probability decays too fast relative to the data.

This in turn begs the question of what these heavier-than-exponential tails imply. The plots suggest that relatively long waiting times are more likely to occur than what is expected from a random process, presumably bookended with increasingly likely clusters of events dispersed amongst smaller waiting times, hence intermittency. When considered from the point of view of survivorship analysis, the plots suggest that the likelihood of "ending" a waiting time decreases with increasing waiting time (relative to that expected for a random process). The longer waiting times presumably reflect temporary upstream "storage" of particles... essentially a preconditioning of bed conditions that yields to avalanching... whose effects tend to be separated in time with small feed rates, then involving increasing overlap as feed rates increase?

From Figure 7, I estimate mean waiting times between emigration events of 2.3, 1.8, 0.75, 0.4 and 0.35 s. Recognizing error in my estimates, these translate to rates of 26, 33, 80, 150 and 171 MPM. The last three are close to the input rates (40, 60, 80, 160 and 200 MPM).

The first two are a mismatch. Do these reflect a mismatch in the (total) time averaged mass balance? Namely, if I understand correctly, the mean waiting time $W_m$ in minutes is $W/60 = 1/f_{output}$, where $f_{output}$ is the number of particles emigrating per minute. Averaged over a long period of time, then with steady conditions, $f_{input} = f_{output}$, so $W \sim 60/f_{input}$. Correct? The forms of the distributions in Figure 6 are unknown. Perhaps calculations of the (changing) average and variance as the time series lengthen might offer a hint whether these moments converge.

**Page 10, Figure 4:** Might I suggest simplifying this figure such that it just shows the unit impulses versus time? As presented, one might be tempted to conclude that an emigration event can involve a fraction of a particle.

**Page 12, Line 6:** How are particle speeds defined? Are these streamwise speeds ($u_p = \Delta x/\Delta t$)? Or speeds involving two dimensions ($\sqrt{u_p^2 + w_p^2}$)? Would this matter?

The phrase "slow speeds ($< 0.1$ mm/s) are associated with essentially immobile particles" implies that all particle speeds are being calculated, not just the speeds of those particles defined previously (Page 8) as being mobile. Is this correct? If so, does the bimodal behavior vanish if only mobile particles are considered?

**Page 17, Figure 10:** I am a bit confused by these figures. First the horizontal axes. In 10A, KE has only positive values, so I assume the definition means that a decrease in particle KE means a positive KE "deposited into the bed," and I assume this represents the KE deposited by individual collisions. In 10B, the horizontal axis spans negative values (although it appears no negative values are plotted), but the magnitude plotted on the positive part of the axis is very different from 10A. So what is the significance of the negative values shown on the axis? According to the caption and the axis labels, this axis (and the magnitudes involves) should be the same as in 10A? Now the vertical axis in 10B. The orientations of the violin plots represent distributions of KE, not of the number of moving particles. But the caption implies that these pertain to the numbers of moving particles — which is the vertical axis. Do the violin plots actually need to be oriented vertically?

Regarding Figure 10A, once a particle is set in motion, presumably there is some hand off to fluid forces, which, together with particle-bed collisions, sets the total travel distance (a random variable) of any individual particle. If so, then this travel distance should not "care" about the KE extracted from the collision that started it, only its own collisions with the bed, which generally tend to shorten its travel distance relative to what would occur in the absence of these collisions. In turn, the "cumulative displacement of all mobilized particles" (which I am assuming is the sum of all travel distances of particles initiated from a collision) must therefore strongly depend on the total "number of particles mobilized," which does care about the KE extracted. It seems that 10A and 10B share information.

Both the KE change with a particle collision and the number of particles mobilized from this collision are random variables. From a statistical mechanics perspective, these define a potentially valuable joint probability distribution. (Each of the violin plots in 10B

then reflects a conditional distribution of the number of mobilized particles for a given KE extraction.) Might it be possible to illustrate some rendition of this joint distribution? And its marginal distributions?

If I correctly understand the information contained in Figure 10, it highlights the idea that neither the KE extraction nor the number of mobilized particles is a deterministic quantity. This also means that, in this problem, the effective coefficient of restitution does not "belong" just to the impacting particle, but rather, it belongs to the particle, the microstructure of the bed, and the associated stability of the bed, perhaps preconditioned by preceding impacts.

djf

---

## Referee Comment (RC2) · C. Ancey (Referee) · 6 Mar 2018

One weakness of extant models of sediment transport is related to collective motion, including collective entrainment of particles from the bed. There is growing evidence that on many if not most occasions, particles move (or are entrained) not in isolation, but as clusters. To the best of my knowledge, most microstructural models (i.e., those inferring the bulk behavior from the local behavior on the particle scale) consider individual events. For instance, in the model developed by Ancey and coworkers (PRE 2006, JFM 2008, JFM 2014, JGR:ES 2015), a central assumption in their Markovian model is that only a single event (e.g., entrainment or deposition) can occur within

a small time increment. Multiple events cannot be considered in jump Markov processes. Correlated motion is another complication, which is poorly accounted for by existing models.

The authors take a first stab at this problem by running experiments in which sediment is replaced with marbles. This allows the authors to reduce the degree of complexity of the problem and look at the mechanisms that drive collective entrainment. I know of no similar study. While there are studies on how particles form clusters when they come to a halt (e.g., Strom et al. JHE 2004), there is little information on collective entrainment. In this respect, the paper is topical. The authors do not present many results, they seem to be skeptical about the chances of ending up with clear ideas on collective entrainment, but the paper is a good starting point. If it does not solve the problem, it should spark interest and lead to further investigations.

I think the paper could be accepted after minor changes. As it stands it suffers from many inaccuracies, mostly of semantic order, which does not make it possible to understand unambiguously what the authors mean. I provide a list of some of the points that jump out at me when reading the paper. I also think that there is an imbalance between the length of the introduction and the subsequent developments. I suggest shortening the introduction and focusing on the key issues in our current understanding of bedload transport. I will take a keen interest in the future results.

Christophe Ancey

Detailed remarks: • Title: it does not seem to reflect the contents of this paper. • Abstract: it should be shortened. Part of the material is not related to what the authors found out. For instance, a sentence such as "A general statistical framework has been developed" led me to think that they developed a theoretical model. Some expressions like "stochastic fluctuations", "probabilistic motion", "relax" sound weird to me. • On p. 2, L5-10: granular avalanches are mass movements (particles move en masse), whereas in many cases, bedload transport involve particles clusters that take the form

of superficial sheets (carpet of moving particles). Therefore, in my opinion, the analogy is not obvious. I agree with the authors that some collective entrainment events may result from avalanches (e.g., on banks or lee slides of bedforms), but this does not seem the only scenario. • P. 2, L14, the citation to Albert (2000) is not related to the statement in this sentence. • P. 2, L27-35: there is a misconception with regards to the term "collective entrainment" used by Ancey et al. Please read [1] more carefully. In my model, I figured out that large fluctuations could be captured by assuming that the probability of entrainment depends on the number of moving particles. That is why the coefficient mu was referred to as the collective entrainment coefficient as it accounts for collective effects in the stream. This does not mean that many particles are entrained at the same time. • P3 L3, I do not think that Einstein mentioned the analogy with Brownian motion. • P4 L1: the sentence may be unclear to many readers. What do the authors mean with "continuously driven limits"? Define "particle activity" (the term was introduced by Furbish et al. recently and not all readers are familiar with this terminology). • P4 L28: remove [] • P8 L11: why do the authors mean with "intermittency"? Intermittent refers to a process that stops and starts. This is the definition used in studies of fluid turbulence, see the related chapter in [2]. Recently, within the bedload transport community, a number of authors have started to use "intermittent" as "fluctuating", but this is not the common definition of this word. Note that for any stochastic process, the time-averaged value converges to a steady state value (if the mean exists, so some heavy tailed distributions are excluded), this is not the signature of a process that would be intermittent • P9 L13: "confounds efforts" -> I understand that the authors failed (or partially managed) to obtain the characteristic time, but then they wrote that they were able to compute it. I am at a loss how to understand this sentence. • P9 L26: a Poisson distribution is a discrete probability distribution whereas the waiting time is a continuous random variable. Did the authors mean "exponential distribution"? • P15 L3: "analogous to avalanches": did they mean that they observed particles moving en masse? • P15 L9: what is D? • Figure 10: "deposited" sounds strange. "imparted" would not be more correct? • P18 L1: "the

similarity of (. . .)" -> I do not understand what the authors mean exactly. Is "depen-dence" correct? • P 19: there are two citations to Ancey et al. (2008). [1] Ancey, C., P. Bohorquez, and J. Heyman, Stochastic interpretation of the advection diffusion equation and its relevance to bed load transport, Journal of Geophysical Research: Earth Surface, 120, 2529-2551, 2015. [2] Frisch, U., Turbulence, Cambridge University Press, Cambridge, 1995.

---

## Author Comment (AC1) · 22 Jun 2018

**Response to Reviewers**

Dylan B. Lee and Douglas J. Jerolmack

20 June, 2018

**1 Response to referee 1 (David Furbish)**

**1.1 Summary**

The Reviewer is generally supportive of the work, and sees how it builds on probabilistic formulations of bed-load transport and links in particular to work on entrainment (and collective entrainment). The Reviewer seeks clarification of some technical points and also some phrasing.

In most cases we have changed the text/phrasing following the suggestions of the Reviewer. We have also clarified some of the more subtle aspects of our analysis and plots to clarify the quantities presented.

**1.2 Response to specific items**

*Page 1, Line 23: Although I appreciate the intention of the phrasing, I suggest rewording, or just deleting the word "unfortunately."*
    We have deleted the word 'unfortunately'.

*Page 2, Line 15: The idea of "hysteresis" used here is unclear. Is this point needed?*
    We agree that hysteresis in the context of bedload transport is still ill-defined and unecessary to the presentation of the current results. We have removed it from the discussion here.

*Page 2, Line 21: Is it correct that the mean hop distance can be calculated from the entrainment probability?*
    This is not correct as the kinematics of hop distances will be governed by flow velocity and grain travel time (Furbish 2012 series of papers). We have deleted this sentence. And replaced it with: "$\bar{L}_x$ is the mean hop length."

*Page 2, Line 32: Does the collective entrainment rate increase, or just its relative contribution, as the transport rate decreases?*

This is a good question. In table 3 of Ancey et al. (J. Fluid Mech. 2008) the estimated collective entrainment rate varies by 1 while the rate of individual entrainment varies by a factor of three indicating that it stays relatively constant. Similarly, Ma et al. (J. Geophys. Res. 2014) shows that the collective entrainment rate could stay constant across the range of transport rates examined in the experiments analyzed in that work. Data from Heyman et al. (J. Geophys. Res. 2016) also indicate that the collective entrainment rate is a constant that determines the rate at which active particles become entrained. Given this, we have revised this line to read: "As the mean transport rate is lowered, the relative contribution of $\mu$ derived from the models must increase in order to reproduce the observed growth in variance of bed-load activity (i.e., the number density of moving grains)."

*Page 3, Line 7: I recommend replacing "exponential" with nonlinear.*
We have followed this recommendation.

*Page 3, Line 9: I recommend replacing "continuum" with "continuum-like," as the rarefied conditions involved in this problem do not satisfy the continuum assumption. The referenced expressions are continuum-like only in that they have the continuous form of continuum equations, when in fact they pertain to ensemble expected conditions.*
We have followed this recommendation.

*Page 4, Line 28: typo present []*
Typo deleted.

*Page 8, Lines 11–19: It is unclear what quantity is being calculated. (I gather later that this is either the activity or the flux, normalized by the average?) What is the total time series length from which the 500 samples are drawn (for specified $\tau$ )? Do these samples overlap (representing the possibility of non-independence)? How might this algorithm differ from calculating the quantity of interest directly from the series with successively increasing duration $\tau$ (a standard approach for examining convergence)?*
$t_{conv}$ is being computed from time series of the observed emigration events. To estimate $t_{conv}$ the quantity being computed is the standard deviation of the emigration rate normalized by the average emigration rate of the bootstrapped samples collected during the $\tau$ in question. We state this in page 8, line 16-18. Emigration series lengths were 316, 1845, 6000, 4966, and 900 seconds for the 200, 160, 80, 60, 40 marbles per minute experiments respectively. For a given $\tau$ the time series is drawn from the emigration series length for that experiment minus the duration of $\tau$. There was a misstatement in the original text as to the resampling. This method is very close to the "standard approach" that the reviewer mentioned; it is perhaps slightly better only in that the bootstrapping procedure potentially removes any bias that might be present in always starting from the beginning of a time series.

*Page 9, Line 9: Were these identified acceleration spikes used in a specific manner later? For example, in the later calculations of kinetic energy changes? If so, are these component accelerations, or computed as changes in trajectory resolved speeds? At 120 fps, I can readily imagine that the acceleration time series are quite noisy, such that identifying impacts involves significant uncertainty?*

The identified acceleration spikes were used to determine the approximate time of impact. The accelerations used in the impact analysis are the total magnitude of the accelerations obtained using the trajectory resolved x and y acceleration componenents. We get the acceleration components by numerical double differentiation of the x and y positions. To resolve this ambiguity have revised the line in question to read: "The x and y components of the saltator trajectories were then numerically differentiated twice to obtain acceleration components which were then converted to magnitudes of grain accelerations through time."

There is significant amplification of noise associated with the double differentation operation. However, the change in speed associated with an impact is dramatic relative to other acceleration events. This enables a high threshold to be used when detecting changes in the acceleration record (we used a threshold of $14.4m/s^2$). An example saltator acceleration record can be seen in figure 1. It is likely that our imposition of a threshold means that we miss some small impacts. In most cases these impactors would be transitional between rolling and saltation whereas it was our goal with this analysis to analyze the impacts of vigorously saltating grains.

Once we have identified when impacts have occured, the portion of the trajectory immediately before and after the identified time of the peak acceleration (the red dots in the example record in figure 1) is then used to estimate the kinetic energy deposited into the bed.

*Page 9, Line 25: The appropriate comparison is with an exponential distribution rather than a Poisson distribution. The latter is a discrete distribution of the (integer) number of events occurring within a specified interval, assuming a stationary Poisson process. The exponential distribution describes the wait times between successive events, assuming a stationary Poisson process. (Is this why the dashed line appears to segmented parts in Figure 6? Where the kinks in the line coincide with integer values >= 100 and the left part of the line is extrapolated toward zero?)... (refer to review for full item text)*

The reviewer is correct, it should be an exponential distribution. We have changed this in the text.

*Page 10, Figure 4: Might I suggest simplifying this figure such that it just shows the unit impulses versus time? As presented, one might be tempted to*

[Figure]

Figure 1: Impact detection for a typical saltator. Panel A shows the saltator trajectory used for the impact detection. There are 4 impacts/rebound events that occur during this trajectory within the viewing window and these have been labeled. Panel B shows the time series of acceleration magnitudes obtained from this trajectory with the estimate of the time at which the 4 impact events highlighted with red dots.

*conclude that an emigration event can involve a fraction of a particle.*

We have edited the figure so that number of particles on y axis are listed soley as a unit impulse.

*Page 12, Line 6: How are particle speeds defined? Are these streamwise speeds ($u_p = \Delta x/\Delta t$)? Or speeds involving two dimensions ($\sqrt{u_p^2 + w_p^2}$) ? Would this matter? The phrase "slow speeds (¡ 0.1 mm/s) are associated with essentially immobile particles" implies that all particle speeds are being calculated, not just the speeds of those particles defined previously (Page 8) as being mobile. Is this correct? If so, does the bimodal behavior vanish if only mobile particles are considered?*

Particle speeds are defined as speeds involving two dimensions ($\sqrt{u_p^2 + w_p^2}$). Figure 2 shows the distribution of streamwise speeds ($u_p = \Delta x/\Delta t$) for the experiments above speeds of .1 mm/s. Above .1 mm/s the qualitative change in the distributions appears to be the same independent of the type of speed used. Yes we calculate the speeds of all particles independent of our mobile/immobile cutoff. Judging from the behavior of the streamwise speed distributions as seen in figure 2 the growth in the bimodality with decreasing feed frequency does not vanish if you consider the more immobile fraction of the bed. The immobile fraction is represented by the left tails of the distribution.

To clarify how particle speeds are defined in the present context we have added the following sentence: "The speed of a given particle is computed as $\sqrt{u_p^2 + w_p^2}$. Where $u_p$ and $w_p$ are the different components of the particle velocity."

*Page 17, Figure 10: I am a bit confused by these figures. First the horizontal axes. In 10A, KE has only positive values, so I assume the definition means that a decrease in particle KE means a positive KE "deposited into the bed," and I assume this represents the KE deposited by individual collisions. In 10B, the horizontal axis spans negative values (although it appears no negative values are plotted), but the magnitude plotted on the positive part of the axis is very different from 10A. So what is the significance of the negative values shown on the axis? According to the caption and the axis labels, this axis (and the magnitudes involves) should be the same as in 10A? Now the vertical axis in 10B. The orientations of the violin plots represent distributions of KE, not of the number of moving particles. But the caption implies that these pertain to the numbers of moving particles — which is the vertical axis. Do the violin plots actually need to be oriented vertically? ... (refer to review for full item text)*

Yes, we consider that the loss in the particle KE is seen as a positive KE input or 'deposition' into the bed. We have clarified this choice of sign for KE by adding the following sentences after page 16 line 10: "The total deposited KE for each event is the sum of the deposited KE associated with each impact observed

[Figure]

Figure 2: A figure showing the distribution of streamwise speeds for the 5 experiments studied.

during that event. Deposited KE is a positive value in all observed impacts, in other words all observed impacts resulted in the impacting particle losing KE into the bed." This modification is also meant to highlight that the total deposited KE for each event is indeed obtained by analyzing the KE deposition associated with individual impacts that occur during an event. It is the total event-deposited KE (or the summed KE deposition observed over all the impacts in the event) that we are plotting in Figure 10. This is why we label the x-axis "KE deposited into bed during event."

The negative values along the 10B horizontal axis do not signify anything. The reviewer is correct that we do not observe impacts with coefficients of restitition greater than 1 so we do not see the bed imparting KE to a colliding grain. The kernel density estimate that provides the smoothed histograms shown in 10B can estimate very low negative values at the end of the tail of some of the distributions with low mean values of deposited KE. This does not have any physcial reality. We have modified Figure 10 so that only positive values are shown 10B. The discrepency in the maximal values is real. This is because as the total number of particles moved per event increases the number of observed events with that number of particles goes down and you can't get good statistics. For example there are 333 observed events where 1 particle is mobile, 206 events with 2 mobile particles and so on. In general this number goes down as you increase the number of particles active during an event. So we capped the distributions plotted in 10B to those where there were at least 8 KE values for which N number of particles moved. By the time there are over 40 particles moving during an event, there aren't very many observed events where this happens. Higher values of KE deposition such as those plotted in 10A are associated with event sizes that don't occur as frequently in these experiments.

The discrepency between the vertical axis in 10B and the figure caption is an error. When we plot the violin plots we are in fact plotting the distribution of KE values needed to move a certain number of particles. So the horizontal orientation of the violin plots is correct but the caption is wrong. We have revised part of the caption text so that this discrepency is corrected.

We should clarify that the cumulative displacement of all mobilized particles represents the sum of the travel distances of all particles initiated from all collisions observed during a transport event as well as the displacement of the impactors themselves, which are also classified as mobile particles. We have clarified this in page 16 line 6 of the draft text by adding the sentence: "Here, the cumulative displacement of all mobilized particles includes the displacement of impacting saltators which are considered to be part of the mobile population."

**2 Response to referee 2 (Christoph Ancey)**

**2.1 Summary**

The Reviewer suggests that the paper can be published with minor revisions. The Reviewer points out that the approach is novel and the observations of collective entrainment from collisions have not been shown before. However, the Reviewer also suggests that there are few results presented, and that there are many phrases or semantic issues that hinder understanding.

We concur that our results represent a good start on this problem — as the Reviewer suggests — but that much more work should be done. That said, significant technical challenges needed to be overcome in order to detect and quantify collective enrainment, and also to relate this behavior to collision dynamics and energetics. We have made changes to the text to clarify and/or justify phrasing in the paper.

**2.2 Response to specific items**

*I suggest shortening the introduction and focusing on the key issues in our current understanding of bedload transport.*

We appreciate the suggestion to make the introduction more focused and parsimomious. We have removed phrases from each paragraph in the introduction to shorten it. We do feel that it is important, however, to keep the structural elements of the introduction intact since they frame our approach to the problem.

*Title: it does not seem to reflect the contents of this paper*

We have changed the title to "Determining the scales of collective entrainment in collision-driven bed load". This seems right since we explicitly measure both the magnitude of collectively entrained grain clusters and also the time between transport events. We assume that the Reviewer objected to the inclusion of "intermittent", based on later comments.

*Abstract: it should be shortened. Part of the material is not related to what the authors found out. For instance, a sentence such as "A general statistical framework has been developed" led me to think that they developed a theoretical model. Some expressions like "stochastic fluctuations", "probabilistic motion", "relax" sound weird to me.*

We have removed the sentence "A general statistical mechanical framework...", which shortens the abstract, clears up the ambiguity raised by the Reviewer, and gets of of "probabilistic motion". We feel that "stochastic fluctuations" is a term that has a general understanding in the community to mean "apparently random variations". The term "relax" is used here in the way it used in physics problems related to avalanche-type dynamics; to describe the

failure, flow and eventual stoppage of a collection of grains.

*On p. 2, L5-10: granular avalanches are mass movements (particles move en masse), whereas in many cases, bedload transport involve particles clusters that take the form of superficial sheets (carpet of moving particles). Therefore, in my opinion, the analogy is not obvious. I agree with the authors that some collective entrainment events may result from avalanches (e.g., on banks or lee slides of bedforms), but this does not seem the only scenario.*

We appreciate the point of view of the Reviewer, and agree that this use of the term "avalanche" can be misleading if one is thinking literally of avalanches (or related landslides). On the other hand, we intend the term avalanche to be understood in the statistical physics sense — bursts of transport that exhibit some kind of spatially-extended correlation. This is useful because there are descriptions of avalanche-like behavior that are applied to a variety of different systems (with different dimensions). We have added a new sentence that clarifies the use of the term avalanche in our paper.

*p. 2, L14, the citation to Albert (2000) is not related to the statement in this sentence.*

Reference removed.

*P. 2, L27-35: there is a misconception with regards to the term "collective entrainment" used by Ancey et al. Please read [1] more carefully. In my model, I figured out that large fluctuations could be captured by assuming that the probability of entrainment depends on the number of moving particles. That is why the coefficient mu was referred to as the collective entrainment coefficient as it accounts for collective effects in the stream. This does not mean that many particles are entrained at the same time.*

Thanks to the Reviewer for this clarification. We realize now that this is *our interpretation of your work*, that multiple particles at once are entrained and that this can explain the fluctuations. We have edited the text to reflect that this view is our hypothesis.

*I do not think that Einstein mentioned the analogy with Brownian motion.*

Removed.

*P4 L1: the sentence may be unclear to many readers. What do the authors mean with "continuously driven limits"? Define "particle activity" (the term was introduced by Furbish et al. recently and not all readers are familiar with this terminology).*

Continuously driven was defined earlier in reference to the Hwa and Kardar

(1992) paper, meaning that transport is steady and smooth. We now define particle activity.

*P4 L28: remove []*
Done.

*P8 L11: why do the authors mean with "intermittency"? Intermittent refers to a process that stops and starts. This is the definition used in studies of fluid turbulence, see the related chapter in [2]. Recently, within the bedload transport community, a number of authors have started to use "intermittent" as "fluctuating", but this is not the common definition of this word. Note that for any stochastic process, the time-averaged value converges to a steady state value (if the mean exists, so some heavy tailed distributions are excluded), this is not the signature of a process that would be intermittent*

We actually mean intermittency in exacty the way the Reviewer does; a process that starts and stops. For us, the time series of particle transport shows intermittency because there are bursts of transport and intervening periods of no transport. We are aware of the turbulence definition as pointed out in Ref [2] by the Reviewer. In turbulent flows of course the flow does not stop and start, so one approach used in that reference is to examine the 4th moment (flatness or kurtosis) of the velocity fluctuations to get a sense of how 'on' and 'off', or bursty, the flow is. But in our system transport truly starts and stops. Note that we sample at a fine enough scale that the emigration number (number of particles passing a line) is binary — it is only zero or 1. Therefore, for our data, the variance of emigration is a function of the amount of switching off and on (i.e., intermittency). We have altered the text to make this point more clear.

*P9 L13: "confounds efforts" -¿ I understand that the authors failed (or partially managed) to obtain the characteristic time, but then they wrote that they were able to compute it. I am at a loss how to understand this sentence.*
Removed sentence.

*P9 L26: a Poisson distribution is a discrete probability distribution whereas the waiting time is a continuous random variable. Did the authors mean "exponential distribution"?*
This has been corrected. See previous response to Reviewer 1, who also pointed this out. We made an error.

*P15 L3: "analogous to avalanches": did they mean that they observed particles moving en masse?*
See comments above. We have now clarified what we mean by avalanche.

*P15 L9: what is D?*
Fixed.

*Figure 10: "deposited" sounds strange. "imparted" would not be more correct?*
Fair enough, though in many studies of collisions of beams of energetic particles, the term 'deposited' is used. We have made our definition more clear so that hopefully there is not confusion.

*P18 L1: "the similarity of (: : :)" -¿ I do not understand what the authors mean exactly. Is "dependence" correct?*
We mean that the scale of collective entrainment — in terms of typical number of particles — does not change. We have clarified this in the text.

*P 19: there are two citations to Ancey et al. (2008).*
Thanks for pointing this out. Corrected.